# Differences in Mother–Infant Bond and Social Behavior of African Elephant Calves Living In Situ and Ex Situ

**DOI:** 10.3390/ani13193051

**Published:** 2023-09-28

**Authors:** Franziska Hoerner, Jake Rendle-Worthington, Arne Lawrenz, Ann-Kathrin Oerke, Karsten Damerau, Santiago Borragán Santos, Therese Hard, Gela Preisfeld

**Affiliations:** 1Department of Zoology, University of Wuppertal, 42119 Wuppertal, Germany; apreis@uni-wuppertal.de; 2eleCREW, Victoria Falls XRH4+72, Zimbabwe; ceo@elecrew.org; 3Zoo Wuppertal, 42117 Wuppertal, Germany; lawrenz@zoo-wuppertal.de; 4Endocrinology Laboratory, German Primate Centre, 37077 Goettingen, Germany; akoerke@dpz.eu; 5Department of Ecology, Europa-Universität Flensburg, 24943 Flensburg, Germany; karsten.damerau@uni-flensburg.de; 6Parque de Cabarceno, 39690 Obregón, Cantabria, Spain; sborragan@cantur.com; 7Boras Djurpark, 506 31 Borås, Sweden; th@knuthenborg.dk

**Keywords:** *Loxodonta africana*, zoo elephants, wild elephants, distance keeping, development, human care

## Abstract

**Simple Summary:**

In the wild, African elephant calves must stay close to their mothers and the family unit as the African environment holds many threats. African elephant calves in zoos are raised in a protected environment. Therefore, we hypothesize that calves ex situ hold bigger distances and behave differently than in situ. Additionally, those differences are likely to increase with further zoo generations. This study used ethological research methods to compare the mother–calf bond of African elephant calves in situ and ex situ (first and second generation). The results showed that ex situ living calves of both generations maintain greater distances to their mothers and show a wider variation (positive and negative) in behavior than in situ. The detected differences indicate that calves ex situ can behave more freely as they are in a protected environment. Therefore, they can develop faster than in the wild, which agrees with similar findings on African elephant calf development and adult African elephants. The hypothesis that differences between in situ and ex situ increase with the zoo generations could not be verified. Hence, modifications in behavior under different environmental selection pressures may be adaptive.

**Abstract:**

African zoo elephants live in safe environments with sufficient resources, are protected from threats, and have their health and body conditions cared for. Calves ex situ undergo the same developmental stages as in situ and are raised by the whole family unit. However, due to environmental differences, there might be behavioral modifications between calves in situ and ex situ. We hypothesize that these differences increase with ongoing generations. This ethological study compares social and general behavior and the distance calves kept to their mothers’ between calves of the first (F1) and second (F2) zoo generation and the wild. Using ethological methods, data were collected for ~90 in situ calves and 16 ex situ (8 F1, 8 F2) between the ages of 0.5 to 4 years (120 observation hours per group). Results showed that in situ calves spent significantly more time close to mothers than the F1 and the F2 zoo generations (F1/in situ: *p* = <0.001; F2/in situ: *p =* 0.007). The behaviors of eating, drinking, trunk movement, washing, and affiliative behaviors showed significant differences between in situ and ex situ calves. The amount and distribution of affiliative and agonistic behavior initiated and received by calves was displayed with a greater variety ex situ. Ex situ calves not only performed affiliative but, in contrast to the in situ, also agonistic behavior (F1/in situ: initiated *p* = 0.002, received *p* = 0.010; F2/in situ: initiated *p* = 0.050, received *p* = 0.037). The comparison of zoo generations suggests that differences did not increase with the generation. The more casual binding between mothers and offspring in zoos and the age-dependent improvement of social behavior of zoo-born calves are seen as a result of elephants’ adaptation to secure zoo conditions. The results of this study agree with the faster development of ex situ African elephants, like earlier puberty and more frequent breeding patterns, as known from the literature.

## 1. Introduction

African elephants (*Loxodonta africana*) are known for their complex and close social bonds. Calves are born into stable families and cared for by their mothers, other females (allomothers), and older siblings [1,2,3]. The death of a mother during the first 24 months of life will leave the calf with almost no chances of survival in situ [4,5,6] and even in captivity [7]. Tactile, visual, olfactory, and acoustic contact between mothers and calves is essential [4,5,6]. This can be seen in data from the Amboseli population, representing the most complete long-term dataset in the demography of wild African elephants [6]. There, calves spend about 56% of their time in close contact with their mothers during the first two years [6]. During this period, the close bond is maintained by both mother and calf, but will loosen after two years and is then pursued more by the mother as the mother is responsible for the calf’s safety and survival [6].

Besides the contact with their mothers, during the first two years, the calves were observed to spend approximately 20% of their time at a physical contact distance to the next family member [6]. Only about 10% of the time they were observed to be more than five meters away from their next neighbor [6]. Charif et al. [8] detected that close spatial bonds are even maintained by adult female elephants of the same bond groups (related family), which were found to have coordinated movement and preserve a distance of no more than 0.5 km for most of their time.

African elephant calves’ development is subdivided into seven stages, as listed in Table 1 [4,5,6,7,8,9,10].

Calves in zoos possibly undergo the same developmental stages and are described by Andrews et al. [11], Webber [12], and Freeman et al. [13]. However, social constellations are different from the wild, and zoo family units are not necessarily related, as in the wild. Due to the increase in captive breeding, zoos are now at the F2 generation (second generation of zoo-born elephants, both parent animals being of the F1 generation), with very few even reaching the F3 generation (third generation of zoo-born elephants) [14]. This aspect is essential as F1 calves (first generation of zoo-born calves) were born and raised by mothers imported from the wild, lacking their mother’s help and assistance. However, F2 calves are already born and raised by mothers who grew up in captivity and often in the presence of their grandmothers, which resembles the family structure from the wild [15].

It has been shown that elephants in zoos reach fertility at a younger age. Females start to show ovarian cycles at 6 to 7 years and can give birth for the first time at 8 to 9 years [16]. While data on the onset of the ovarian cycle in wild females are missing, the first births are reported in cows mostly between 12 and 16 years of age, the earliest being reported at 9 years old (± one month) [9,10,17,18]. Males in zoos must not show musth to be able to breed [19] and can sire offspring as young as 9 to 10 years old [19]. Whereas in the wild, males reach musth for the first time around the age of 12–14 years and were observed to be accepted as mating partners by cows only at the age of 25 years [10].

If we are to suppose that African elephants in zoos reach puberty much earlier than in the wild, in that case, it is possible that African elephant calves in zoos also develop faster than those in the wild, and most likely recreate faster. Preliminary data were collected by Hoerner et al. [under review], who found that calves living ex situ tended to maintain greater distances to their mothers than reported for their conspecifics in situ as known from the literature [4,6,8]. Calves in zoos were observed to spend up to 31% of their time at a distance of more than five meters from their mothers already at the age of three days [20,21,22]. This was observed in male and female calves and of the matriarch and sub-dominant cows. This spatial detachment was observed to increase with the age of the calves [20,21,22], whereas it is unclear if this feature will increase with future generations.

The new generations in the zoos (F2 and, most recently, F3) are no longer solely socialized by wild-born elephants but by zoo-born elephants. Additionally, the import of wild elephants is considered outdated [23]. Therefore, in situ-born elephants become less represented in zoos. Calves adapt the social and behavioral patterns of the relatives that raise them [9,24,25]. This results in our two hypotheses: (1) Captive elephant calves keep bigger distances to their mothers and show different social and general behavior than wild elephant calves. (2) Those differences between wild and captive elephant calves increase with the next zoo generations.

We tackled those hypotheses by combining ethological research in situ and ex situ to learn about possible differences in the behavior and distance keeping of calves brought up correspondingly. To investigate the second hypothesis, we collected data for ex situ calves from the F1 and F2 generations.

## 2. Material and Methods

### 2.1. Animals

Data were collected for a total number of ~106 elephant calves of three different groups: (I) F1 generation ex situ, (II) F2 generation ex situ, (III) in situ. Within Europe, data were collected in four zoos for 16 African elephants—8 of the F1 generation and 8 of the F2 generation. Calves from zoos had between 240 and 1315 m^2^/elephant of space. A possible impact of the varying enclosure size on the calves’ behavior was eliminated in a previous study [22].

The animals observed in situ were selected to reflect the same age group and distribution as the F1 and F2 groups. The age of in situ calves was estimated by body size and confirmed by knowledge of local rangers. Calves of all data sets were further sampled by age group according to developmental stages [4,5,6]. Data for this group were collected in the Jafuta Reserve, the Zambezi National Park, and Hwange National Park in Zimbabwe. The areas in which family units were observed were sparse miombo woodland with nearby water sources. As no register for the family units in the observation areas was available, it could not reliably be determined whether the same families were observed on several occasions. Family units were followed by vehicles. The age limit for the sample animals was drawn at approximately four years, as gender-related differences in behavior become significant from that age [6]. Family units of calves observed in situ had varying sizes between 4 and 26 animals, comprising family units with older daughter elephants, subadult males, and allomothers.

Table 2 displays the information of the animals, valid for the time of data collection.

### 2.2. Ethological Data Collection

The behavior of the calves was measured utilizing an ethogram extracted from Poole & Granli [26,27], listing 16 behavioral categories (Table 3), and two research methods: the Social Distance Method, Focal Animal Sampling [28,29,30].

The Social Distance Method measured the distance between calves and mothers, dividing the distance into five parameters: *tactile contact*, <1 m, 1–3 m, 3–5 m, and >5 m [28,30,31]. The distance was noted every 60 s using continuous sampling. This parameter was used to analyze the mother–calf relationship [6].

Utilizing the ethogram and Focal Animal Sampling, the general behavior of the calves was observed. These data were used to generate a behavioral profile of the calves of different origins [28,30,31,32,33]. Here again, the interval for data registration was 60 s using continuous recording.

Additionally, the calves’ social behavior within the family units was measured. Therefore, all affiliative and agonistic contact, either initiated or received by the calves, were measured. These data were collected using continuous recording [28,30,31,32,33].

Data collection in zoos took place between 2016 and 2021 and in the wild from February to March 2023. Each calf of the F1 and F2 ex situ generation was observed for 15 h, resulting in an observation time of 120 h for each sample group. For the in situ sample group, the observation time was also 120 h. However, here, calves were not observed for 15 h but ~3–4 h each, as the single individuals could not be tracked again reliably. Data in situ and ex situ were only collected when animals could behave freely without human interactions. During in situ data collection, observers held a big distance from the family units (at least 200 m). They used binoculars for a better view, ensuring that calves and family units were not influenced in their behavior by human presence.

### 2.3. Data Analysis

For data analysis, all data sets were classified numerically by summing up all data to a joined maximum of 100% [34,35,36]. For statistical analysis data for calves were additionally sampled according to age group [4,5,6]. As all sample groups were chosen to be age matched in sample size, we did not include age as a variable. Statistical analysis was performed with SPSS version 29 (IBM SPSS Statistics 29). All data sets were tested for distribution with the Shapiro–Wilk test and the Kolmogorov–Smirnov test [37]. As neither of the tests resulted in a homogenous or even distribution of the data sets, a graphical analysis of the Q-Q plots was used. All data sets were identified as non-parametric [38,39]. The Therefore, the Kruskal-Wallis calculation for non-parametric datasets was used to detected significant differences between the three sample groups (F1 ex situ, F2 ex situ, and in situ). In case of significant differences a Post hoc test was calculated to detect which sample groups showed differences [40,41]. For all calculations, the level of significance was set at *p ≤* 0.05 (normal significance) and *p ≤* 0.001 (strong significance) [42,43].

## 3. Results

### 3.1. Distance to Mother

Statistical data analysis on the distance between calves and their mothers with the Kruskal–Wallis test detected significant differences. We found that in situ living calves predominantly spend time in tactile contact with their mothers (M = 58.11%), in comparison to the ex situ calves of the F1 (M = 7.22%) and F2 generation (M = 15.27%) (Figure 1). Calves of the F1 and F2 zoo generations spent the majority of time in the distance category *< 1 m* to their mothers (F1: M = 35.05%, F2: M = 32.50%). It can also be seen that calves living in the wild barely spend time at a distance further than 1 m from their mothers.

Comparing the data of calves of the F1 zoo generation and the wild with the Post hoc test, significant differences were detected in the distance categories *tactile*, *1–3 m, 3–5 m,* and *>5 m*. The difference between the F1 zoo generation and wild calves was not significant for the distance category *<1 m*. Calves of the F2 zoo generation and the wild also showed significant differences for the categories *tactile*, *1–3 m, 3–5 m, and >5 m*. The comparison of the data on distance to the mother did not detect any significant differences between calves of the F1 and F2 zoo generations (see Table 4).

### 3.2. General Behavior

Calves of the F1 and F2 zoo generation mostly displayed the behavior *eating* (F1: M = 39.703, F2: M = 35.13), followed by the behavior *affiliative contact* (F1: M = 12.435, F2: M = 14.944) (Figure 2). In situ calves, on the other hand, mostly showed *affiliative contact* (M = 48.22). No other behavior was recorded as often as the three categories mentioned above.

The statistical analysis (Table 5) of the amount the behavioral categories were shown by calves of the F1 and F2 generations ex situ and living in situ (see Table 3) revealed significant differences between calves of the F1 generation and in situ, and also of the F2 generation and in situ in the five behavioral categories *eat* (in situ < ex situ), *drink* (in situ > ex situ), *trunk movement* (in situ < ex situ), *wash* (in situ > ex situ) and *affiliative behavior* (in situ > ex situ). For the behavioral categories *suckle*, *walk*, *run*, *wash*, *sleep*, *social play*, *lone play*, *escape*, *seeking rescue*, *rescuing*, and *threatening*, no significant differences were found between in situ and F1 ex situ, as well as in situ and F2 ex situ. Significant differences between the F1 and F2 generations were only detected for the behavioral category *suckle* (F1 < F2).

### 3.3. Social Behavior

The boxplots in Figure 3 demonstrate that calves of the F1 zoo generation initiated affiliative behavior of M = 91.41% and agonistic behavior of M = 8.59%. They received affiliative behavior of M = 83.65% and agonistic behavior of M = 13.35%. Calves of the F2 zoo generation initiated affiliative behavior of M = 94.61% and agonistic behavior of M = 5.39%. They received affiliative behavior of M = 85.78% and agonistic behavior of M = 14.22%. Calves living in the wild received and sent only affiliative behavior. The behavior initiated by in situ calves was 100% affiliative with no agonistic behavior. Additionally, they received affiliative behavior of M = 98.73%, with a single outlier.

The analysis of the social behavior of the calves of the three test groups also showed significant differences between calves living ex situ and in situ (Table 6). There were no significant differences between the F1 and F2 zoo generations. Significances were detected between the F1 generation and in situ calves in all four social behavior categories. Significant differences were found in received affiliative and agonistic behavior between the F2 generation and calves in situ.

## 4. Discussion

The results of the distances between calves and their mothers confirm the first hypothesis that there are significant differences between African elephant calves living in the wild and zoo environments. Calves in situ spending a majority of their time at a very close spatial distance to their mothers (*tactile* and *<1 m*: M = 92.78%), which was significantly higher than for the F1 zoo generation (*tactile* and *<1 m:* M = 42.27%) and the F2 zoo generation (*tactile* and *<1 m:* M = 46.78%), agrees with former observations of Webber [12]. She states that in situ calves stay almost continuously at a close spatial distance to their mothers and that this is not valid for calves born in zoos. However, while comparing African and Asian calves in situ and ex situ, Webber only found this difference in Asian elephant calves living in zoos. Berg [44] also observed captive African elephant calves up to six months of age and observed that they spend 70–75% of their time in body contact with other individuals. The data at hand first observed a spatial detachment for African calves living ex situ. A possible explanation for this spatial detachment between mothers and calves living in a zoo environment is the absence of possible threats (predators, losing the family unit, lack of water). In the wild, a close spatial bond with the mother elephant is crucial for the calf’s survival [2,6,45].

The second hypothesis of this study, stating that those differences might increase with the F2 zoo generation, was not confirmed by data on the distance kept by calves from their mothers. No significant differences between the distance keeping of the F1 and F2 generations were detected. The significance level even decreases from strong to normal with the generations. We interpret the increasing spatial detachment between calves and mothers observed in zoos not as an issue of concern regarding elephant breeding in zoos. Previous studies on adult F1 ex situ generation elephants detected species-specific social behavior and bonds that subsist over years and generations [46,47].

The data for the 16 general behavior categories for calves living in situ collected in this study resemble those described by other researchers [2,6,48], as do the data for the calves from zoos [12,49,50,51].

Other than the data on the distances between calves and mothers, the data on the general behavior of the calves do not display as many significant differences between the in situ and ex situ calves of both generations. The calves of the sample group in situ were observed to spend significantly more time drinking and washing. The observation spot can explain this, as data in situ were frequently collected close to a water hole when families moved out of dense bushes and could be easily observed. Hence, calves spent more time drinking and bathing on those occasions.

Calves in situ spent less time eating than calves of both generations ex situ. A possible reason is that animals must feel safe eating [2]. In the wild, calves were observed to be more anxious than in the zoos and, therefore, might spend less time eating than calves in zoos, which are constantly in a safe environment. Additionally, elephants in zoos have access to food almost continually and can eat without stress and fear [52,53]. In situ, elephants must feed in the open bush or grassland [17]. Ex situ calves of this study had constant access to food in the form of hay, branches, and occasionally fruits and vegetables. However, during our observation in the wild, calves also had constant access to food, such as grass and branches. As observations were made in March and April, the vegetation was dense due to the rainy season.

The behavioral category, *trunk movement*, was displayed significantly less frequently by calves living in the wild. A possible reason for calves living in zoos displaying this behavior more frequently is that calves in an ex situ environment have more time to train their trunk, instead of concentrating on following the mother and the family. It can be assumed that the protected environment leads to quicker development, as can also be noted by an earlier start of breeding [16,19]. The more leisure behavior observed from the calves in this study agrees with Webber et al.’s interpretations [54]. They observed that ex situ African and Asian elephant calves spend more time playing than in situ calves. They also conclude that this difference originates from the more peaceful zoo environment that gives calves more opportunities for playing behavior [54]. Another possible explanation for this difference is that calves in situ have other occupations besides playing. In the wild, they must gain ecological and social knowledge and specific skills to ensure survival [4,5,6,9]. This is not required in zoos.

The significantly higher affiliative behavior displayed by calves from the wild compared to the F1 and F2 generations in zoos indicates differing social behavior for calves living in these different environments. A possible reason for this is the ever-changing presence of other elephants, independent of the family, in the wild. Zoo elephants live in generally stable family units that change less frequently than in the wild. The number of changes in nearby animals likely impacts the affiliative behavior of calves in the wild, which depend on the care and positive reactions of other elephants and, therefore, almost solely displaying affiliative behavior [2,15].

Similar to the data for the distance between mother and calf, data on the general behavior of the calves showed no trend of an increase in the differences between ex situ and in situ calves with the next generation of zoo elephants.

The amount of affiliative behavior initiated and received by calves from the zoo in this study is significantly lower than that in the wild, where they are known to be treated with intense care and affiliative behavior by family members [2,17,45,55,56]. Also, the distribution of affiliative and agonistic behavior between ex situ and in situ calves differs in this study, as the in situ sample group was barely observed to initiate or receive agonistic behavior. Nevertheless, the ex situ calves also initiated and received significantly more affiliative than agonistic behavior, corresponding to the wild’s social behavior [2,45,55]. A possible reason for the lower amount of social behavior recorded for zoo calves is the enrichment and safety that the zoo environment supplies. While in the wild, calves must stay close, follow, and be in contact with their mothers almost constantly [2,17,45,55], a safe zoo environment allows them to devote themselves to other activities. This also enhances the faster development of elephants in zoos [16,19].

Also, affiliative behavior is less crucial for calves living in zoo environments as they live in rarely changing social groups. Many young animals display more affiliative than agonistic behavior in wild environments [57].

While many behavioral patterns of in situ and ex situ calves differ significantly in the study at hand, studies on adult African elephants of the F1 generation detected species-specific social behavior, with a strong majority of affiliative behavior initiated and received by family members, as stated before [46,47]. Hence, the question of whether differences in social behavior between ex situ and in situ living African elephants increase with the generations cannot be answered in this paper.

## 5. Conclusions

Despite the varying sample sizes and observation hours per animal, the present study found significant differences in the distance keeping and the general and social behavior of in situ and ex situ African elephant calves to their mothers and other family units or family members. These findings agree with former findings on Asian elephant calves in zoos by Webber [12]. However, they did not make the same observations for African elephant calves [12].

Calves living in a safe environment are not hesitant to separate earlier from their mothers as this involves less risk for them. Ex situ calves are less hesitant to contact other elephants with agonistic behavior. Additionally, instead of following their mothers and keeping social contact with the family members like in the wild, calves in a safe zoo environment have more time to observe, learn, play, adapt social behavior, eat, and compete and, therefore, can develop quicker. This faster development of ex situ calves corresponds with the earlier maturity and breeding of zoo elephants [16,19]. African elephant calves ex situ are more independent than in the wild and spend more time eating and interacting with others, following the faster growth rate and the general pattern of enhanced development rates ex situ [16,19].

If zoos continue to breed elephants to generate a self-sustaining population—which is necessary, as the import of wild elephants is considered outdated [23]—it needs to be ensured by ethological research that the elephants bred and socialized there show (social) behavior that does not indicate a negative impact on their wellbeing. Social interactions and touch in captive elephant calves are highly relevant during early development and are associated with prosocial behavior and elephant welfare [13]. Therefore, falsifying hypothesis two is essential as this is reassuring for the ex situ breeding program which seeks to establish an independent stock of zoo elephants living under the best welfare conditions [19].

## Figures and Tables

**Figure 1 animals-13-03051-f001:**
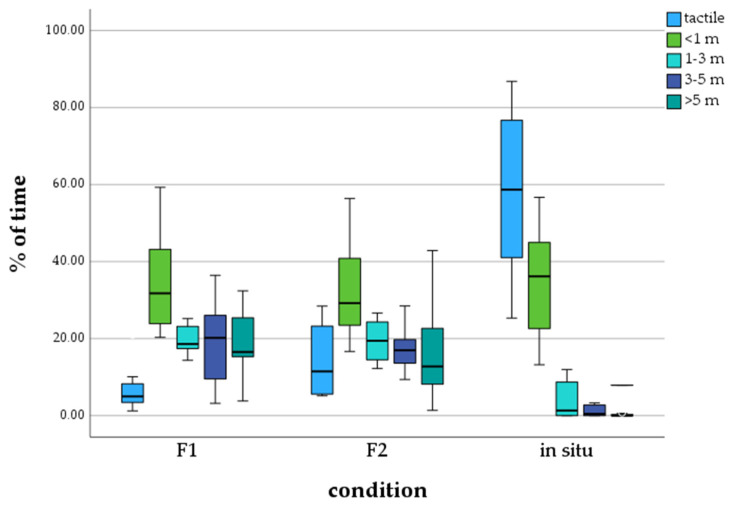
Percentage of time calves spend in a distance category to mother, depending on generation/environment.

**Figure 2 animals-13-03051-f002:**
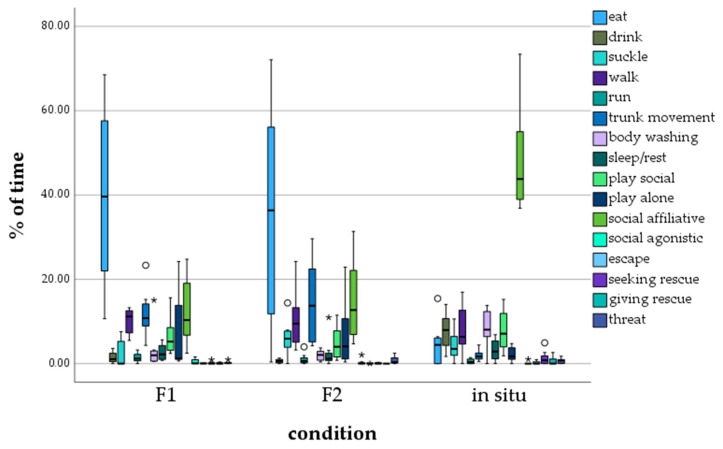
Percentage of time calves showed a certain behavior, depending on generation/environment. ° = outlier, * = extreme outlier.

**Figure 3 animals-13-03051-f003:**
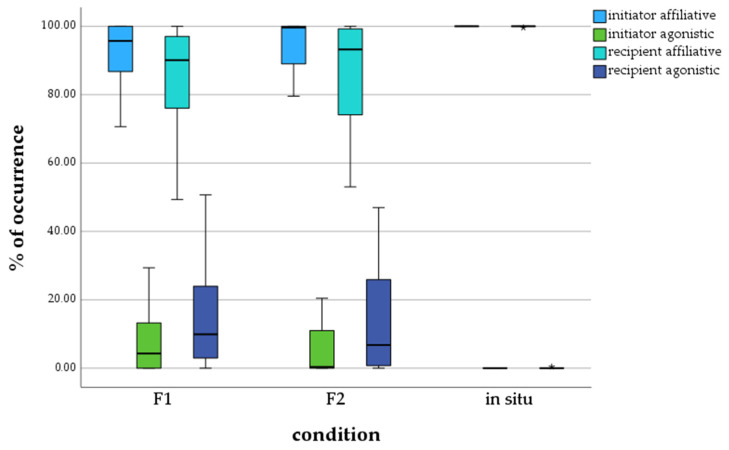
Percentage of affiliative and agonistic behavior initiated and received by calves, depending on generation/environment. * = extreme outlier.

**Table 1 animals-13-03051-t001:** Developmental stages of elephant calves.

**Age**	**Development**
0–6 months	- Learn how to walk stable - Learn how to use the trunk for suckling
7–12 months	- Learn how to use the trunk for foraging - Main nutrition shift from milk to solid food
13–24 months	- First decrease in contact with mothers - Pick up on playing behavior
25–36 months	- Start to show more agonistic behavior, which is relevant for learning to compete in rivalries - Increase in gender-specific differences - Peak in playing behavior (way of learning social behavior)
37–48 months	- Classified as youngsters - Gender-specific differences in social behavior become more significant
49–60 months	- Social play (that is, among other functions, intended to prepare youngsters for breeding behavior), such as climbing and chasing others, increases significantly
60-plus months	- Classified as young adults - Males and females start to become fertile in situ, even though it still takes several years for them (especially the males) to mate successfully - Behavior shifts even stronger toward behavior related to breeding - Young bulls might already have left the natal family to socialize with bachelor groups

**Table 2 animals-13-03051-t002:** List of elephants.

Generation/Origin	Elephant	Sex	Year of Birth	Age at Data Collection	Number of Playmates	Total Number
F1/ex situ	Ts	M	2020	6 months	3	8
	Ku	M	2021	6 months	3	
	Gu	M	2019	12 months	3	
	Tu	F	2016	24 months	3	
	Jo	M	2014	36 months	3	
	Maj	F	2017	48 months	4	
	Ch	F	2017	48 months	3	
	Sa	F	2017	48 months	3	
F2/ex situ	Ki	F	2020	6 months	4	8
	Ne	M	2021	6 months	1	
	El	F	2019	12 months	2	
	Mar	F	2019	30 months	2	
	Tora	M	2018	36 months	4	
	Tori	M	2018	42 months	4	
	Ta	F	2016	48 months	2	
	Ay	M	2016	48 months	1	
in situ	-	M & F	-	3–48 months	2–13	~90

**Table 3 animals-13-03051-t003:** Ethogram for the data collection on the general behavior of calves (extracted from Poole & Granli [26,27]).

Label	Behavior
Eat	Eating food using the trunk
Drink	Drinking water using the trunk
Suckle	Suckling milk from the mother’s breast
Walk	Walking at a slow pace, no more than one step per s, with the purpose to go somewhere
Run	Running in an enhanced space, more than one step per s, to run away from something or get somewhere fast for safety
Trunk movement	Moving either the tip of the trunk or the whole trunk for practice or to search the ground for food/objects
Washing	Washing the body with mud/water, sand bathing, rubbing the body on something to clean the skin, and protecting from mosquitos and sun
Sleep	Sleeping or resting in a lying or standing position with the eyes closed
Social play	Playing with one or more other individuals
Lone play	Playing individually with oneself or an object
Affiliative behavior	Behaving positively with other individuals (e.g., touching with the trunk, helping behavior)
Agonistic behavior	Behaving negatively with other individuals (e.g., pushing with trunk, tusk, or body)
Escaping	Running from something while showing signs of fear (screaming, low tail, head high)
Seeking rescue	Running towards other individuals in fear (e.g., screaming, low tail, head high) and hiding under/behind them for protection
Rescuing	Standing over/in front of other individuals for protection, after that individual ran towards them to seek rescue (see above)
Threatening	Pacing towards something, head, trunk, and ears high, sometimes trumpeting

**Table 4 animals-13-03051-t004:** Kruskal–Wallis calculation and Post hoc test for the position of calves to their mothers, depending on generation/environment.

Kruskal–Wallis			Tactile	<1 m	1–3 m	3–5 m	>5 m
Kruskal–Wallis H			15.495	0.222	15.082	13.113	13.995
df			2	2	2	2	2
Asymp. Sig.			<0.001	0.895	<.001	0.001	<0.001
Monte Carlo Sig.	Sig.		<0.001	0.897	<.001	<0.001	<0.001
	99% Confidence Interval	Lower Bound	0.000	0.889	0.000	0.000	0.000
		Upper Bound	0.000	0.905	0.000	0.001	0.001
Post hoc							
Sig.		F1/F2	0.224	0.651	0.819	0.887	0.570
		F1/in situ	<0.001	0.895	<0.001	0.002	<0.001
		F2/in situ	0.007	0.740	0.001	0.002	0.003

**Table 5 animals-13-03051-t005:** Kruskal–Wallis calculation and Post hoc test for the general behavior of calves, depending on generation/environment.

Kruskal–Wallis			Eat	Drink	Suckle	Walk	Run	Trunk Move.	Wash	Sleep	Social Play	Lone Play	Affiliative	Agonistic	Escape	Seek Resc.	Rescue	Threat
M	F1		39.703	1.459	2.264	10.105	1.415	11.924	3.335	2.679	6.495	7.121	12.435	0.453	0.058	0.164	0.104	0.236
	F2		35.13	0.616	6.179	10.398	1.048	14.56	1.955	2.47	4.88	6.869	14.944	0.31125	0.004	0.088	0.0	0.778
	in situ		4.566	7.713	4.3	8.032	0.536	1.939	8.431	3.206	7.884	2.225	48.22	0.13	0.225	1.273	0.606	0.653
Kruskal–Wallis H			12.026	14.443	3.879	0.945	2.720	14.495	6.335	1.165	1.565	0.665	15.405	3.569	2.658	4.648	3.063	1.682
df			2	2	2	2	2	2	2	2	2	2	2	2	2	2	2	2
Asymp. Sig.			0.002	<0.001	0.144	0.623	0.257	<0.001	0.042	0.559	0.457	0.717	<0.001	0.168	0.265	0.098	0.216	0.431
M.C. Sig.	Sig.		0.001	<0.001	0.147	0.647	0.267	<0.001	0.039	0.580	0.474	0.733	<0.001	0.156	0.279	0.101	0.313	0.447
	99% C.I.	L.B.	0.000	0.000	0.138	0.635	0.256	0.000	0.034	0.567	0.461	0.722	0.000	0.147	0.267	0.093	0.301	0.434
		U.B.	0.002	0.000	0.156	0.660	0.278	0.000	0.043	0.593	0.487	0.744	0.000	0.165	0.290	0.109	0.325	0.460
Post hoc																		
Sig.	F1/F2		0.620	0.339	0.049	0.750	0.327	0.777	0.860	0.447	0.437	0.646	0.832	0.214	0.103	0.921	0.100	0.269
	F1/in situ		0.001	0.007	0.299	0.340	0.101	0.002	0.037	0.777	0.646	0.724	<0.001	0.064	0.454	0.056	0.765	0.254
	F2/in situ		0.006	<0.001	0.352	0.525	0.510	<0.001	0.024	0.297	0.216	0.416	0.001	0.541	0.379	0.069	0.179	0.972

**Table 6 animals-13-03051-t006:** Kruskal–Wallis calculation and Post hoc test for agonistic and affiliative behavior as initiator and recipient for calves, depending on generation/environment.

Kruskal–Wallis			Affiliative Initiator	Affiliative Recipient	Agonistic Initiator	Agonistic Recipient
Kruskal–Wallis H			6.105	6.105	7.581	7.581
df			2	2	2	2
Asymp. Sig.			0.046	0.044	0.018	0.019
Monte Carlo Sig.	Sig.		0.046	0.044	0.018	0.019
	99% Confidence Interval	Lower Bound	0.041	0.039	0.015	0.016
		Upper Bound	0.051	0.049	0.022	0.023
Post hoc						
Sig.		F1/F2	0.649	0.576	0.649	0.576
		F1/in situ	0.022	0.010	0.022	0.010
		F2/in situ	0.050	0.037	0.050	0.037

## Data Availability

Please contact franziska.hoerner@uni-wuppertal.de for data.

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
