# Peer review of "Differences in Mother–Infant Bond and Social Behavior of African Elephant Calves Living In Situ and Ex Situ"

_animals, 2023, doi:10.3390/ani13193051_

Round 1
Reviewer 1 Report
Comments on Horner et al.
A paper of relevance to zoo biologists, as well as evidencing the developmental plasticity in long-lived species.
There are a few odd uses of terminology which could be corrected throughout.
1. Title. Please use either infant of calf. Comparative context = infant (NOT child)
2. Elephants live in family units. Herd is a livestock term for a grouping. Unrelated to the complex structure of elephants. Change throughout.
Typos in summary: mothers (plural) not possessive
Introduction, line 57: “pursued by mother = mother is responsible for….
Line 82: play is not specific to these functions. Please restate as “among other functions, breeding”
96: no furthermore here…. Just start the sentence
Lee et al. 2018 report earliest first birth at 9 years old (± one month).
If you could not identify the families, their matriarch or the calves from observation to observation in the wild, how could you determine their ages? Need these details in the methods. How habituated were these wild elephants. Could the behaviours you observed have been influenced by the presence of an observer? I note the high contact frequency (ma-inf figure) and suspect that the elephants were observed in very specific contexts (e.g. during rest, suckling, at water? )
Why is “lone play” called neutral. It’s self simulation plus a variety of locomotory activities (practice) and can be object play.
Line 188: what do you mean “other than”? By comparison to?
Table 4 and text. These comparisons are meaningless unless the age of each of the subjects has been controlled for in the statistical comparisons.
264: note Webber only found a statistical difference (controlling for age of calf) in captive Asian elephants, but she compared and tested African wild and African captive too. Yours in NOT the first such study (line 266)
You could add one line to the implications:
That elephants in captivity are more independent than those in the wild, and spend more time interacting and eating is in accordance with the more rapid growth rates and general pattern of enhanced developmental rates. An age-controlled comparison is needed to validate these patterns?
Some minor edits suggested to improve clarity.
Author Response
Dear Reviewer,
thank you very much for your review and comments on our manuscript. We found them very helpful and edited the manuscript accordingly. We are looking forward to your reply.
All the best,
Franziska Hörner
- Please use either infant of calf. Comparative context = infant (NOT child) – edited accordingly.
- Elephants live in family units. Herd is a livestock term for a grouping. Unrelated to the complex structure of elephants. Change throughout. – edited accordingly.
- Typos in summary: mothers (plural) not possessive – edited accordingly.
- Introduction, line 57: “pursued by mother = mother is responsible for…. – we edited that sentence: “During this period, the close bond is maintained by both mother and calf but will loosen after two years and is then pursued more by the mother, as the mother is responsible for the calfs’ savety and survival [6].”
- Line 82: play is not specific to these functions. Please restate as “among other functions, breeding” – edited accordingly.
- 96: no furthermore here…. Just start the sentence – edited accordingly.
- Lee et al. 2018 report earliest first birth at 9 years old (± one month). – Lee et al 2016 stated that first birth is most frequently between 12-16 years, 9 years is the exception. We clarified that in the manuscript.
- If you could not identify the families, their matriarch or the calves from observation to observation in the wild, how could you determine their ages? Need these details in the methods. – We clarified this information: “The age of in situ calves was estimated by the body-size of the calves and confirmed by knowledge of local rangers.”
- How habituated were these wild elephants. Could the behaviours you observed have been influenced by the presence of an observer? I note the high contact frequency (ma-inf figure) and suspect that the elephants were observed in very specific contexts (e.g. during rest, suckling, at water? ) – The calves/herds were observed in a big variety of situations and locations. When coming to a waterhole, while resting, while feeding, while walking, while playing… We made sure that the presence of the observer did not influence the families, as we held a big distance (at least 200 meters) and mostly used binoculars to get a better view. We clarified that in the manuscript: “During in situ data collection, observers held big distance to the family units (at least 200 meters) and used binoculars for a better view, ensuring that calves and family units were not influenced in their behavior by human presence.”
- Why is “lone play” called neutral. It’s self simulation plus a variety of locomotory activities (practice) and can be object play. – We referred to this as neutral, as it does not contain social (hence affiliative or agonistic) interaction. However, we see how our phrasing can be misleading. We edited that.
- Line 188: what do you mean “other than”? By comparison to? – Yes, we meant to say “in comparison to”. We edited this.
- Table 4 and text. These comparisons are meaningless unless the age of each of the subjects has been controlled for in the statistical comparisons. – In the statistical analysis we considered the age groups (as listed in the introduction) and made analysis according to that. Also, we ensured that there was no significant difference between the age distribution of the three sample groups. We clarified that in the manuscript.
- 264: note Webber only found a statistical difference (controlling for age of calf) in captive Asian elephants, but she compared and tested African wild and African captive too. Yours in NOT the first such study (line 266) – Thanks for pointing that out. We clarified that in the manuscript. We meant to say that out study is the first to find that difference in African calves, not the first such study. “However, while comparing African and Asian calves in situ and ex situ, Webber only made this finding in Asian elephant calves living in zoos.”
- You could add one line to the implications:
That elephants in captivity are more independent than those in the wild, and spend more time interacting and eating is in accordance with the more rapid growth rates and general pattern of enhanced developmental rates. An age-controlled comparison is needed to validate these patterns? – Thank you for the input! We edited the section accordingly. Age was considered in the statistical analysis.

Reviewer 2 Report

The English language could be improved but mainly the focus should be on the scientific writing.
Author Response
Dear Reviewer,
Thank you very much for your review and comments on our manuscript. We found them very helpful and edited the manuscript accordingly. We are looking forward to your reply.
All the best,
Franziska Hörner
Material and Methods
- Line 153-154 Table 2: The definitions of the behaviours are in need for improvement on the details. – We amended the Table to be more specific.
- For example, the difference in how to record Walk or Run is only described as a difference in speed. One way to describe this is to write how many feet that are in contact with ground at any one point. – edited.
- When elephants Sleep, are their eyes closed? Nor all animals close their eyes in sleep. – edited.
- Helping behaviour how does that look like? Seeking rescue/Rescuing standing behind other individuals. Is that at a certain distance? If I see one elephant standing behind another when I see a herd, should that always be recorded as Seeking rescue? – edited.
- Line 162: , the interval for data registration You have used Focal-Animal-Sampling but not previously stated if you used continuous recording (which is the most usual combination with focal animal sampling) or instantaneous sampling. Clarify what you use and if you record behaviours with interval sampling or One-zero sampling, if instantaneous sampling is your method. – Thanks for pointing that out. We used continuous recording. We added that information.
- Line 166: This data were collected as libitum. The method ad libitum is not an appropriate method if you want to answer specific questions in ethology. The method is described as one where you take notes on different behaviours that you see and that seems relevant at the time. Usually used when establishing an ethogram for a species of for the purpose of pilot observations before the study. From your description in the paragraph, I think that you have used continuous recording of all affiliative/agonistic behaviours could that be it? – You are right, thank you for pointing that out. We recorded the behavior continuously during the observation time.
- Line 174: all data sets were classified numerically.. This need more description. If you register a distance every 60s and present it as percent of time, you need to clarify how this calculation was made. This is also connected to how you recorded your data, if you used one-zero sampling time percent of time is not an appropriate way to describe your data. – We clarified that: “For data analysis, all data sets were classified numerically by summing up all data of a data set to a joined maximum of 100 % [34-36].”
Results
- Line 194-195 Figure 1: Is the data presented in figure 1 per calf? Since the groups are so very different in size this is important to know. And also, how was the age distribution the in-situ group? For F1 and F2 the distribution is well described in Table 1 but not for the in-situ group. If the young ages are over represented in that group in might explain the high number of recordings of tactile that you see. – We added information on the age groups. In situ calves were chosen and sampled to match the sample groups and age distributions of the ex situ Statistical analysis was facilitated considering the age groups.
- Line 213: No other behavioural category was recorded for M<15%... What do you mean with this sentence? - No other behaviour was recorded as often as the three categories above mentioned. We rephrased that.
- Line 219: the amount of specific behavioural categories What behavioural categories? You have not specified in the M&M section what categories of behaviours you have. Be more precise in your wording. – We edited this.
- Line 226: Significances between You mean differences??? – We meant significant differences. We clarified that.
- Line 238: Rephrase the sentence. If you have a result that the animal received affiliative behaviour 98.73% of the recorded social behaviours, that is still a high number and using only imply something else. – We edited this.
Discussion
- Line 273-275: This sentence needs rephrasing. – edited.
- Line 337-341: This statement is very strong considering that you have only recorded the behaviour for two generations ex-situ. We can t neglect the possibility of changes in social behaviour for future generations in captivity only based on these results. – Thanks for pointing this out. We edited this paragraph.
Conclusions
- Line 343: differences in the physical. It is not a good wording here. Change it to general behaviour/distance since it is what you have observed. – We rephrased that.
- You make a lot of statements in your conclusions that are not supported by the results from your study. I give some examples below:
- Line 357-360: Moderate your conclusion – We erased this paragraph.
- Line 365-369: This you can t say at all from your results. You can t for see what happens in future ex-situ populations. – We erased this paragraph.

Reviewer 3 Report
line 44 Later development? line 61 rewrite. 67-72 cut, start with At the age of 9 months Lee and Moss etc etc. 74. Ex and in situ is confusing, why not use zoo bred/ captive and wild?
Table 2. Definition of Moving Trunk, does this mean the end of the trunk the whole trunk and if so by how much?
Play: is this Object play or self-play or social play?
"Rescuing and protecting" ? How do you know? Why not just stand over and later explain why you think this.
The amount of space in the zoos may be very important for their activities but there is no mention of this. (267) Why less eating in the wild when they have to learn what to eat, where to find it, how to get it and eat it?? Also the captive born usually have concentrate food as well as browse: did they have browse available all the time and if so how many species/ bits of plant etc all of which will affect time eating. (291). They ate more perhaps because they had nothing else to do and obesity is a problem in zoos too!
306 more play in zoos what type of play? Perhaps as LESS to do not more to do! In the wild the elephants have to become good ecologists as well as sociologists and they have a lot to learn quickly, not so in the zoo necessarily where all provided. The "security" post is valid, but there are many different other concerns here too.
Fig 3 is not clear. I would expect the wild calves to be affiliative not aggressive as they are learning the social contract which is vital to them, and less important to captive elephants in rarely changing social groups. Young animals of many species show more affiliation that aggression when they are in wild type environments. (Journal of Consciousness Exploration & Research | January 2011 | Vol. 2 | Issue 1 | pp. 119-159 119 Kiley-Worthington, M. A Comparative Study of Equine and Elephant Mental Attributes Leading to an Acceptance of Their Subjectivity & Consciousness, Kiley-Worthington M, J Anim Res Vet Sci 2019, 3: 012 DOI: 10.24966/ARVS-3751/100012. Communication in elephants).
More eating in zoos because less milk dependent is also possible.
295 Yes but have to learn what and how to eat it particularly in wild where more changing, also what time of year ? May be not much to eat that the infants can find... time spent eating depends on availability to a degree in herbivores.
300 Training trunk with play objects? But surely more different objects in the wild, push pull, manipulate to eat etc etc
335. They have a different social contract because of their lifetime experiences, this does not mean that the zoo born elephants will manage well if re-introduced to the wild, so you need to mention this. Also maternal behaviour learning from mother has been shown in several species, particularly humans and cattle ( references if you want).
I think a more id depth discussion would improve this paper which is interesting and has possible application to welfare concerns, but needs the mentioned critics addressed.
Overall the paper is interesting and important, however there are some minor English changes that are needed, and other explanations for some of the conclusions that should be considered.
Details of critique as follows shown to authors and editors.
Author Response
Dear Reviewer,
Thank you very much for your review and comments on our manuscript. We found them very helpful and edited the manuscript accordingly. We are looking forward to your reply.
All the best,
Franziska Hörner
- line 44 Later development? – That was a mistake. We rephrased that into “faster”
- line 61 rewrite. – was edited.
- 67-72 cut, start with At the age of 9 months Lee and Moss etc etc. – We edited this paragraph, however, the seven stages need to stay listed, as we applied those stages to sample our elephants in the different age groups.
- Ex and in situ is confusing, why not use zoo bred/ captive and wild? - Those are the scientific designations for this.
- Table 2. Definition of Moving Trunk, does this mean the end of the trunk the whole trunk and if so by how much? - was clarified: “Moving either the tip or the whole trunk for practice or to search the ground for food/objects”
- Play: is this Object play or self-play or social play? – Was clarified.
- "Rescuing and protecting" ? How do you know? Why not just stand over and later explain why you think this. – Was clarified.
- The amount of space in the zoos may be very important for their activities but there is no mention of this. – Information was added: “Claves from zoos had between 240 and 1,315 m2/elephant of space. A possible impact of the varying enclosure size on the calves’ behavior was eliminated in a previous study [22].”
- (267) Why less eating in the wild when they have to learn what to eat, where to find it, how to get it and eat it?? - because in situ food is not constantly available like in the zoos. We clarified that in the manuscript.
- Also the captive born usually have concentrate food as well as browse: did they have browse available all the time and if so how many species/ bits of plant etc all of which will affect time eating. – we clarified that.
- (291). They ate more perhaps because they had nothing else to do and obesity is a problem in zoos too! – That is true. However, none of the calves observed in this study was obese out of a veterinarian perspective.
- 306 more play in zoos what type of play? Perhaps as LESS to do not more to do! In the wild the elephants have to become good ecologists as well as sociologists and they have a lot to learn quickly, not so in the zoo necessarily where all provided. The "security" post is valid, but there are many different other concerns here too. – Thanks for pointing this out. We edited this paragraph.
- Fig 3 is not clear. I would expect the wild calves to be affiliative not aggressive as they are learning the social contract which is vital to them, and less important to captive elephants in rarely changing social groups. Young animals of many species show more affiliation that aggression when they are in wild type environments. (Journal of Consciousness Exploration & Research | January 2011 | Vol. 2 | Issue 1 | pp. 119-159 119 Kiley-Worthington, M. A Comparative Study of Equine and Elephant Mental Attributes Leading to an Acceptance of Their Subjectivity & Consciousness, Kiley-Worthington M, J Anim Res Vet Sci 2019, 3: 012 DOI: 10.24966/ARVS-3751/100012 . Communication in elephants). – Thanks a lot for pointing this out! Also thank you very much for the source, that is very helpful. We edited the section accordingly.
- More eating in zoos because less milk dependent is also possible. – That would mean that they would spend less time suckling milk ex situ. However, our data does not suggest that.
- 295 Yes but have to learn what and how to eat it particularly in wild where more changing, also what time of year ? May be not much to eat that the infants can find... time spent eating depends on availability to a degree in herbivores. – We clarified that: “Ex situ calves of this study had constant access to food in form of hay, branches and occasionally fruits and vegetables. However, during our observation in the wild, calves also had constant access to food, such as grass and branches. As observations were made in March and April, after the rain-season, the vegetation was dense.”
- 300 Training trunk with play objects? But surely more different objects in the wild, push pull, manipulate to eat etc etc – That is right. We edited that.
- They have a different social contract because of their lifetime experiences, this does not mean that the zoo born elephants will manage well if re-introduced to the wild, so you need to mention this. Also maternal behaviour learning from mother has been shown in several species, particularly humans and cattle ( references if you want). – I am not sure that you are actually referring to line 335. Can you clarify that? Otherwise, I agree with your point. We would be very happy if you could name the references.

Round 2
Reviewer 1 Report
While it is excellent that you have noted that age groups were compared, I see no tests of age in the statistics as presented in the tables etc. WHAT if any was the influence of age on these figures and in the captive, wild comparisons. Please simply state where and how age effects were controlled for. You included age in tests HOW and WHERE? What were the outcomes.
OR - all samples were age matched in sample size so we did not include age as a variable?
Somehow this must be stated.
Otherwise, I think this is a much better version.
Author Response
Dear Reviewer,
Thank you for your response. We edited the section "Data analysis" according to your comment on the age distribution of the sample groups. We chose sample groups to be age matched. Thanks for pointing this out.
Best regards,
Franziska Hörner

Reviewer 2 Report
I'm happy with the changes that have been made since I last read the paper.
Author Response
Dear Reviewer,
Thank you for your response. We are happy, that you approve the edited manuscript. Attached you can find the lasted version.
Best regards,
Franziska Hörner
